# AdaPM: a Partial Momentum Algorithm for LLM Training

## Abstract

In the training of large language models, momentum is often able to achieve significant acceleration and is widely used. However, storing momentum typically presents memory challenges. In this paper, we propose AdaPM, a training strategy that leverages the adaptive partial momentum to implement a memory-efficient optimizer using partial momentum. We first show that the momentum in transformers optimizer also remain highly redundant and demonstrate that most blocks do not require full momentum acceleration, therefore assigning different momentum designs to different blocks. In order to mitigate the high bias and performance loss caused by partial momentum, we further improve the partial momentum by a bias-corrected approach using error-feedback technique. Empirically, we verify that our approach reduces memory by up to over 90% in momentum while maintaining both efficiency and performance for pretraining various language models sized from 60M to 1.5B and for supervised fine-tuning and RLHF. AdaPM can further reduces memory by up to 95% in optimizer states by combining the memory-efficient technique on the secondorder statistic, saving over 30% GPU hours for pretraining GPT-2 1.5B.

## 1 Introduction

Efficient optimizers are a key factor in the success of modern large language models (LLMs)(Vaswani et al., 2017)(Achiam et al., 2023)(Touvron et al., 2023)(Liu et al., 2024). Besides how to accelerate training speed, a major challenge for these optimizers arise from the significant memory overhead they introduce to the limited memory resources. For instance, the Adam optimizer (Kingma & Ba, 2015) requires two additional sets of values, the first and second statistics estimators, for every parameter, which greatly increases the demand on device memory. As a result, the optimizer states alone can rival or even exceed the storage cost of the model parameters themselves. The high memory cost often forces practitioners to employ techniques like CPU offloading and optimizer state sharding (Rajbhandari et al., 2020), which typically trade off the critical training throughput to accommodate training (Rajbhandari et al., 2021).

Therefore, reducing memory consumption not only breaks the threshold to broadens participation in machine learning research, but it also enables larger batch sizes, reduced communication overhead, and thereby directly accelerates optimization.

Current approaches targeting at reducing the optimizer state primarily targets second-order statistics, while comparatively less attention has been devoted to the first-order statistics. The asymmetry lies in their distinct roles. The second-order statistics are non-negative and serve as per-parameter scale estimators. (Zhang et al., 2024b) reveals that they typically exhibit near-uniform scales within architectural units like blocks or neurons. Such redundancy has motivated methods like Adafactor (Shazeer & Stern, 2018) Adam-mini (Zhang et al., 2024b) that greatly compress second-order states. By contrast, the signed first-order statistics encodes update direction (Chen et al., 2023), where approximation error can misalign the ground-truth descent direction (Kunstner et al., 2023)(Fu et al., 2023). Attempts with reduced first-order statistics, such as low-rank update algorithms in (Cosson et al., 2023)(Zhao et al., 2024), experience performance degradation in retraining. Consequently, whether the first-order state admits redundancy as the second-order state and can be substantially compressed without degrading performance remains as an open question (Kunstner et al., 2023).

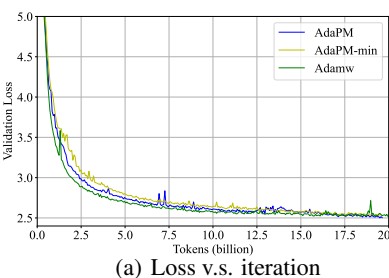 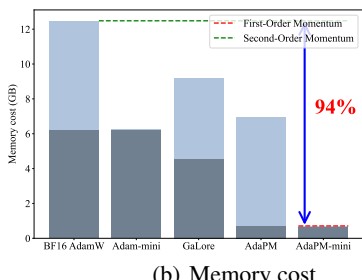

(a) Loss v.s. iteration    (b) Memory cost

Figure 1: AdaPM takes less memory and can reach higher throughput with on par or better performance than AdamW. (a) Results for GPT-2 1.5B pre-training. (b)The memory cost when training GPT-2 1.5B with various optimizers.

We propose the method, **Ada**ptive **P**artial **M**omentum, AdaPM, which reveals that the second-order statistics is also significantly reducible without performance degradation. One first insight originates from the recent analyses that break down the integrated transformer architecture and reveal the heterogeneity across its components. First, the momentum of some components can be completely redundant due to the its nagative effect of enlarging variance. In parallel, the component-wise analyses reveal the components that accommodate low-rank approximations. Besides the component-wise analysis, our second contribution in AdaPM is a residual compensation mechanism that corrects the discrepancy between the full momentum and its low-rank approximation. This mechanism restores the discarded descent directions using the current residual with a rescaling and mitigates the degradation of approximation. Finally, our method is capable of integrating approaches that reduce second-order statistics, such as Adam-mini, thereby allowing the optimizer state's memory footprint to be substantially reduced.

Our experiments demonstrate that AdaPM matches or exceeds the performance of AdamW across various task with only $5\%$ of the momentum is required. We pertaining AdaPM under a GPT-2 series and LLaMA series from 124M to 1.5B and also test AdaPM over Llama-3 8B post-training tasks. AdaPM achieves a consistent comparable or better convergence performance, showcasing strong scalability and robustness across model sizes. Notably, as shown in Fig 1, when combined Adam-mini, AdaPM can even achieves a remarkable $94\%$ memory saving on the optimizer state without sacrificing convergence of AdamW. Moreover, due to higher throughput and larger batch sizes resulting from memory reduction, AdaPM saving over $30\%$ GPU hours for pretraining.

To summarize our contribution:

- Partition approach of Transformers. We explore the effects of blocks in Transformer on momentum acceleration. We find that most blocks do not require full momentum acceleration: (1) Embedding and attn.proj blocks not need momentum acceleration; (2) query, key and mlp needs low rank momentum; (3) value need full momentum.

- Bias-corrected estimation of low-rank momentum. We propose a debias method that achieves unbiased estimation of full momentum which only requires the low-rank momentum with merely $5\%$of the original dimensions.

- Mermory-efficient optimizer. Our work integrates the above partition principle with debiased low-rank momentum estimation, proposing a novel algorithmic framework that significantly reduces memory overhead in Adam optimization. By combining with methods like Adam-mini, this framework achieves over 95% memory savings.

## 2 RELATED WORKS (PRELIMINARY)

**Lightweight optimizers focused on second-order statistics**. Several lightweight optimization algorithms have been developed to reduce computational and memory costs by leveraging second-order statistical information. Adafactor (Shazeer & Stern, 2018) and its variant CAME (Luo et al., 2023) are memory-efficient variant of Adam employing nonnegative low-rank factorization over

Adam's $v$. SM3 (Anil et al., 2019) employs a candidate set, which is derived from the maximum squared gradient within a specific group of parameters defined by a cover of the parameters and determines the learning rate for the i-th parameter by selecting the smallest value from the candidate set. Adam-mini (Zhang et al., 2024b) partitions the parameters into blocks a proposed principle on Hessian structure and assigns a single learning rate for each block using the average of Adam's v in that block.

**Lightweight optimizers focused on subspace learning**. Recent research has shown that the core learning process largely takes place within a much lower-dimensional subspace of the parameter space (Gur-Ari et al., 2018). Research like (Gooneratne et al., 2020)(Yang et al., 2023) applied the low-rank property of gradient during training of neural networks to reduce memory footprint during training. Similar approach has been widely used in meta-learning and continual learning (Lee & Choi, 2018)(Chaudhry et al., 2020). GaLore (Zhao et al., 2024) and a novel variant, Golore (He et al., 2024) calculate a low-rank gradient estimator $\hat{g}$ and then calculates $m$ and $v$ based on this $\hat{g}$.

## 3 STARTING POINT OF MOMENTUM REDUCTION

In this section we present the detail discussion on the potential redundancy of the momentum. We present the theoretical verification an empirical evidence in Section 3.1 and Section 3.2, respectively.

### 3.1 THEORETICAL JUSTIFICATION

Our first illustration of redundancy in momentum is by posing the following fundamental question: Does adding momentum in optimization consistently lowers the validation loss? To investigate, we adopt the standard framework of minimizing the validation risk $\mathcal{R}(\mathbf{W}) = \mathbb{E}_{\mathbf{x},y\sim\mathcal{D}}\ell(\mathbf{W}; (\mathbf{x}, y))$. Let $\mathbf{W}^* \in \arg\min_{\mathbf{W}} \mathcal{R}(\mathbf{W})$ denote the oracle minimizer, and let $\hat{\mathbf{W}}_{\text{opt}}$ denote the output of the optimization algorithm with randomness driven by stochastic gradients.

The distribution of outputs produced by an optimization algorithm exhibits distinct systematic (bias) and stochastic (variance) effects,

The distribution of the outputs produced by an optimization algorithm is comprised by the expectation the algorithm and how stochastic gradients drive the algorithm deviates from the this, yielding the following nature decomposition of the validation loss:

$$\mathcal{R}(\hat{\mathbf{W}}_{\text{opt}}) - \mathcal{R}(\mathbf{W}^*) = \underbrace{\mathcal{R}(\hat{\mathbf{W}}_{\text{opt}}) - \mathcal{R}(\bar{\mathbf{W}}_{\text{opt}})}_{\text{term } \mathcal{A}} + \underbrace{\mathcal{R}(\bar{\mathbf{W}}_{\text{opt}}) - \mathcal{R}(\mathbf{W}^*)}_{\text{term } \mathcal{B}}, \tag{1}$$

where $\bar{\mathbf{W}}_{\text{opt}} = \mathbb{E}\left[\hat{\mathbf{W}}_{\text{opt}}\right]$ is expectation of the algorithm's output.

Introducing momentum into deterministic problems accelerates the optimization, indicating that the term $\mathcal{B}$ can be optimized more efficiently with the addition of momentum. For the term $\mathcal{A}$, momentum offers no guarantee of reducing the negative effect from the gradient randomness; rather, analyses suggest adding momentum may even augment the variance of $\hat{\mathbf{W}}_{\text{opt}}$.

We compare the vanilla SGD without momentum and the accelerated SGD with momentum $1 - \beta$. Smaller $\beta$ corresponds to higher momentum, and when $\beta = 1$, accelerated SGD recovers the vanilla one. The comparison is listed in the following theorem.

**Theorem 1** (Validation Loss Rates for SGD and Accelerated SGD, adapted from Theorem 1 in Zhang et al. (2024a)). *Set a constant stepsize of $\eta = \Theta(1)$ and the number of iteration $T$. After $T$ iterations, the validation loss of vanilla SGD is bounded by $\tilde{\mathcal{O}}\left(T^{1/a-1} + T^{1/a-b/a}\right)$. For the accelerated SGD method with momentum $1 - \beta$ (where $\beta \in [0, 1]$), the validation loss after $T$ iterations is bounded by $\tilde{\mathcal{O}}\left(T^{1/a-1}\beta^{1/a^2-1/a} + T^{1/a-b/a}\beta^{(1/a^2-1/a)(1-b)}\right)$.*

In the context of validation loss, the term $T^{1/a-1}$ in vanilla SGD and $T^{1/a-1}\beta^{1/a^2-1/a}$ in accelerated SGD correspond to term $\mathcal{A}$ in equation 1, reflecting the validation loss increase due to the solution's variance. In parallel, $T^{1/a-b/a}$ in vanilla SGD and $T^{-b/a+1/a}\left(\beta^{1/a^2-a/1}\right)^{-b+1}$ in accelerated SGD capture the deterministic optimization component of the loss and corresponds to $\mathcal{B}$.

As $a, b \geq 1$ Higher momentum with smaller $\beta$ enlarges $\mathcal{A}$. Therefore, when problems can be efficiently optimized by a deterministic non-accelerated algorithm, variance $\mathcal{A}$ dominates and adding momentum is redundant. As a specific case, when $a = b = 2$, the validation loss scales as $\tilde{\mathcal{O}}\left(T^{-1/2}\beta^{-1/4} + T^{-1/2}\beta^{1/4}\right)$, which is minimizes at $\beta = 1$ (no momentum).

## 3.2 Empirical Insights

The above analyses establish a general theoretical principle on the potential redundancy of momentum. In what follows, we turn to presenting empirical observations from transformer training. with insights into potential redundancy of the momentum.

**Sparse Gradients.** One empirical property of the transformer is the sparsity in gradient matrices. Its existence is demonstrated as in Fig 6, where we illustrate the scales of the gradient of the embedding layers and attention projection layers. Fig 6 demonstrates that most of the column/rows of the gradient matrices are filled by near-zero values.

We will demonstrate that the sparse gradients will mitigate the efficacy of the momentum, thereby making the momentum potentially redundant. The logic of sparse gradient's effect on momentum can be summarized as: Low frequency of the gradient signals disrupts gradient accumulation across iterations since most parameter coordinates exhibit gradient signals only rarely, with substantial iteration gaps between occurrences.

**Gradients Concentrate on a Low-Rank Structure.** The second observation leading to the momentum redundancy is a consistent tendency of gradients concentrating on a low-rank subspace during LLM training (Gur-Ari et al., 2018). Besides, the low-rank subspace also displays temporal stability, as the dominant singular directions associated with the low-rank structure remain largely invariant over time. These observations are illustrated in Figure 2, which shows the energy of the momentum in Query, Key and MLP blocks are concentrated in the top $5\%$ eigenvalues.

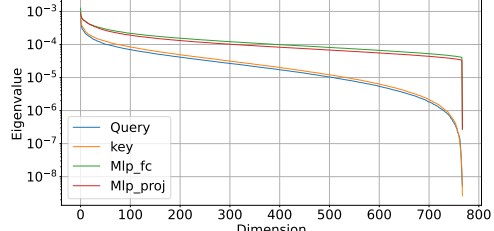

Figure 2: The spectral distribution of features in each block of 10th layer in GPT-2 124M at $10\%$ of the training steps.

The low-rank structure has inspired a class of memory-efficient optimization methods, such as LoRA and its variants (Zhao et al., 2024; Hu et al., 2022; Lialin et al., 2023), which restrict parameter updates to a low-rank subspace while discarding residual components. However, during pretraining, these methods typically underperform their full-rank Adam counterparts. Consequently, this raises the open problem that: Is it possible to attain the performance of full-rank methods while reducing the memory into the low-rank structure?

## 4 Our Method

Claims in Section 3 suggest the potential redundancy of momentum in optimization, while in transformer optimization this redundancy appears less evident. Transformer optimization is neither a easier problem where momentum can be entirely omitted, nor have existing methods effectively exploited low-rank structures to reduce the redundancy in momentum without degrading performance.

This tension can be addressed by the insight that: heterogeneous transformer blocks are better suited to distinct momentum designs. With this insight, we propose our method AdaPM. In AdaPM, instead of treating block with a uniform low-rank design, we introduce a non-uniform momentum design for the transformer blocks, tailored to the optimization difficulty. This includes full momentum, no momentum, and low-rank momentum. This will be detailed in Section 4.1. In parallel, for low-rank momentum, we specifically propose a debiased estimator which empirically **m**itigates the performance degradation caused by compressing gradients into low-rank structures.

Table 1: Comparsion of validation loss with various partition principles on GPT-2 124M. Here Q, K, V, O, Em, stand for Query, Key, Value, and Attention Output, Embedding, respectively, and Full, Low-rank, None stand for Full Momentum, Low-rank Momentum with our correction method, No Momentum, respectively.

| Full | Low-rank | None | Final Loss | Full | Low-rank | None | Final Loss |
|------|----------|------|-----------|------|----------|------|-----------|
| All blocks | - | - | 2.962 | - | Q,K,V, Mlp | Em, O | 3.143($\uparrow$) |
| Q,K,V, Mlp | - | Em, O | 2.964 | V | Q,K, Mlp | Em, O | 2.997 |

## 4.1 NON-UNIFORM MOMENTUM DESIGN

The first component of AdaPM, non-uniform momentum reduction, classifies the reliance on momentum in each transformer block into three distinct regimes:

**(1) Embedding and Attention Output Projection Blocks.** Gradients in these transformer components are sparse and lack temporal persistence, rendering momentum of limited value as discussed in Section 3.2. We disable momentum and use an AdaGrad-style update to reduce the momentum overhead.

**(2) Query, Key, and MLP blocks.** These blocks possess more challenging optimization landscapes. Although momentum accelerates convergence, the gradient signal concentrates on a low-rank subspace leaving the residual with limited information. We therefore adopt a debiased low-rank approximation: (i) compress the momentum via a low-rank projection to suppress redundancy; (ii) correct the induced bias by reintroducing the current-iteration residual. This reduces redundancy while preserving information that naive truncation would discard, thereby maintaining performance. We will detail this debiased estimator in Section 4.2.

**(3) Value Blocks.** For the value layers, however, low-rank updates prove inadequate. For value blocks, our method preserves full momentum update to ensure effective optimization.

Table 1 justifies the division strategy: omitting momentum in the attention and feedforward components leads to a clear degradation in performance; moreover, applying low-rank approximation to the value block—unlike its effect on other blocks—noticeably slows down convergence.

AdaPM achieves substantial momentum reduction from full momentum method like Adam. Table 2 reports the reduction ratio relative to full momentum: a aggregated reduction to merely $5\%$ of the original momentum is guaranteed across models with multiple scales.

## 4.2 DEBIASED LOW-RANK ESTIMATOR

For layers utilizing the low-rank momentum, to avoid performance degradation of discarding the components outside low-rank components, we propose a bias-corrected estimation of low-rank momentum to incorporate residual information. Concretely, our low-rank momentum estimation involves two components: low-rank momentum approximation tracking and a bias-correction step. We first summarize our method for low-rank update in Algorithm 1.

The component of tracking low-rank momentum approximations avoid constraining the approximation to a fixed or rarely updated subspace. We incrementally update a low-rank approximation $\mathbf{L}_t \mathbf{R}_t$ of the momentum $\mathbf{M}_t \in \mathbb{R}^{m \times n}$ in each step. Concretely, at iteration $t$, given parameter matrix $\mathbf{W}_t \in \mathbb{R}^{m \times n}$ and the stochastic gradient $\nabla f(\mathbf{W}_t, \xi_t)$, leveraging the estimate from the previous iteration, $\mathbf{L}_{t-1} \mathbf{R}_{t-1}$, the update is defined by the following optimization problem:

$$\mathbf{L}_t \mathbf{R}_t \in \arg\min_{\mathbf{L}, \mathbf{R}} \left\| \mathbf{L}\mathbf{R} - \left( (1 - \beta_1) \tilde{\nabla} f(W_t) + \beta_1 \mathbf{L}_{t-1} \mathbf{R}_{t-1} \right) \right\|^2. \tag{2}$$

The optimization problem in equation 2 is a standard matrix factorization task, which can be efficiently solved using gradient-based methods. In implementation, We apply gradient descent and warm-start from the previous estimate $\mathbf{L}_{t-1} \mathbf{R}_{t-1}$. The method typically stabilizes within 10 iterations, yielding accurate low-rank momentum updates with negligible overhead.

---

**Algorithm 1** Low Rank Update with Correction for a $m \times n$ layer $\mathbf{W}$

---

**Require:** Weight-decay coefficient $\lambda$, decay rates of momentum $\beta_1, \beta_2$, rank of the momentum approximation matrices $r$ and learning rate schedule $\{\eta_t\}_{t=1}^T$

1:  Initialize $\mathbf{L}_0 \in \mathbb{R}^{m \times r} \leftarrow \mathbf{0}$, $\mathbf{R}_0 \in \mathbb{R}^{r \times n} \leftarrow \mathbf{0}$, $\mathbf{v}_0 \in \mathbb{R}^{m \times n} \leftarrow \mathbf{0}$ and step $t \leftarrow 0$
2:  **for** $t = 1$ **to** $T$ **do**
3:      Obtain mini-batch gradient $\nabla f(\mathbf{W}_t, \xi_t)$
4:      $\mathbf{m}_t \leftarrow (1 - \beta_1)\nabla f(\mathbf{x}_t, \boldsymbol{\xi}_t) + \beta_1 \mathbf{L}_{t-1}\mathbf{R}_{t-1}$
5:      $\mathbf{L}_t, \mathbf{R}_t = \arg\min_{\mathbf{L},\mathbf{R}} \|\mathbf{L}\mathbf{R} - \mathbf{m}_t\|_F^2$
6:      $r_t = \mathbf{m}_t - \mathbf{L}_t\mathbf{R}_t$          ▷ Approximation residual
7:      $\mathbf{v}_t = \beta_2 \mathbf{v}_{t-1} + (1 - \beta_2)[\nabla f(\mathbf{W}_t, \boldsymbol{\xi}_t)]^{\odot 2}$   ▷ Standard second-order momentum update
8:      $\mathbf{m}_t^c = \mathbf{m}_t - \frac{r_t}{1 - \beta_1}$        ▷ Bias correction for low-rank momentum
9:      $\mathbf{W}_{t+1} = \mathbf{W}_t - \eta_t \left( \text{clip}\left( \frac{\mathbf{m}_t^c}{\sqrt{\mathbf{v}_t} + \epsilon}, 1 \right) + \lambda \mathbf{x}_t \right)$
10: **end for**

---

The second component of our method is the bias correction. Directly applying $\mathbf{L}_t\mathbf{R}_t$ as the momentum discards the residual components outside the low-rank structure. Our compensation for the low-rank structure leverages the following one-step residual

$$r_t = \mathbf{L}_t\mathbf{R}_t - \left( (1 - \beta_1)\tilde{\nabla} f(\mathbf{W}_t) + \beta_1 \mathbf{L}_t\mathbf{R}_t \right), \tag{3}$$

At iteration $t$, $r_t$ denotes the approximation error. Because the $\mathbf{L_{t-1}R_{t-1}}$ in the momentum accumulation 2 carries forward the residuals from previous steps. Therefore, the bias accumulates as a weighted sum of past errors $r_{t-1}, r_{t-2}, \cdots$. To compensate for the accumulated bias, we refine the momentum estimate by

$$\mathbf{m}_t^c = \mathbf{m}_t - \frac{r_t}{1 - \beta_1}. \tag{4}$$

where $\frac{r_t}{1-\beta_1}$ serves the residual correction to the low-rank approximation. To justify the correction term $\frac{r_t}{1-\beta_1}$ in equation 4, we assume that the per-iteration residuals are (approximately) stationary.

**Assumption 2** (Stationary Residuals). *The one-step residuals $\{r_t\}_{t \geq 1}$ are identically distributed across iterations, i.e., $r_t \overset{d}{=} r_{t'}$ for all $t, t' \geq 1$.*

The near-stationarity of $r_t$ arises from the smoothing induced by the moving average and the incremental updates of $\mathbf{L}, \mathbf{R}$. Moreover, in practice it suffices that residuals are nearly stationary over short horizons $r_t, r_{t+1}, \ldots, r_{t+k}$, since the exponential moving average down-weights older terms. Empirically, this stationary is illustrated as in Figure 4(b), where we observe high consistency of $r_t$s' distributions over windows of 20 steps.

We compare the compensated momentum equation 4 with the following full-rank momentum without low-rank approximation

$$\mathbf{m}_t^f = (1 - \beta_1)\mathbf{m}_{t-1}^f + \beta_1 \tilde{\nabla} f(\mathbf{W}_t) \quad \text{with} \quad \mathbf{m}_0^f = \mathbf{0}.$$

Under Assumption 2, the following theorem establishes that the proposed debiased momentum precisely eliminates the bias induced by the low rank structure.

**Theorem 3.** *If Assumption 2 holds, then the compensated momentum $\mathbf{m}_t^c$ will asymptotically eliminate the bias in the low-rank estimator $\mathbf{m}_t^f$:*

$$\lim_{t \to \infty} \mathbb{E}[\mathbf{m}_t^c - \mathbf{m}_t^f] = 0.$$

## 5 EXPERIMENT

We are now validating the effectiveness of Adapm on both pre-training and fine-tuning tasks. All GPT-2-1.5B experiments were trained on NVIDIA H800-80GB GPUs, while all other models were trained on NVIDIA A6000 GPUs.

Table 2: Memory cost of AdamW v.s. Adapm. Calculation is based on float32, which is a standard choice for optimizer states.

| Llama-350M | | | GPT-2-1.5B | | |
|---|---|---|---|---|---|
| Algorithm | Memory | GPU Hours | Algorithm | Memory | GPU Hours |
| Adam | 2.72G | 13.8 | Adam | 12.48G | 26.67 |
| Adam-mini | 1.36G | 11.7 | Adam-mini | 6.24G | 20.32 |
| AdaPM | 1.48G ($\downarrow$ **46**%) | 11.9($\downarrow$ **14**%) | AdaPM | 6.98G ($\downarrow$ **44**%) | 22.11 ($\downarrow$ **17**%) |
| AdaPM-mini | 0.12G ($\downarrow$ **96**%) | 9.8($\downarrow$ **29**%) | AdaPM-mini | 0.74G ($\downarrow$ **94**%) | 17.92 ($\downarrow$ **33**%) |

## 5.1 PRETRAINING

**Setups.** We perform pre-training on the GPT-2 series (Brown et al., 2020) (125M to 1.5B parameters) on the OpenWebText (Gokaslan et al., 2019) dataset using the nanoGPT implementation. Following standard setting in Adam-mini (Zhang et al., 2024b), the models are trained with a consistent configuration of 512 batch size, 1024 sequence length and 0.1 weight decay. We pretrain train Llama series (130M to 340M) (Touvron et al., 2023) on C4 (Raffel et al., 2020). For all pretraining cases, we apply a cosine learning rate decay (2000 warm-up steps) and global gradient clipping at 1.0 threshold. We tune the learning rates for all methods and report the curve with the smallest final loss. For Adapm, we constantly set $r = 5\%$ and $T = 100$, and the ablation study can be found in Section 5.1.2. In addition to state-of-art algorithm AdamW, our evaluation compares Adapm with several widely-used memory-efficient optimizers:

- Adafactor (Shazeer & Stern, 2018): We incorporate momentum with $\beta_1 = 0.9$ to ensure a fair comparison with other methods. We apply Adafactor with the default hyperparameters: clipping threshold $d = 1.0$, epsilons =(None, 0.001), $\tau = -0.8$. By tuning the hyperparameters, we set the learning rate as 0.01.

- Galore (Zhao et al., 2024): We set subspace frequency $T$ to 200 and scale factor $\alpha$ to 0.25 across all model sizes. We pick the same rank $r = 0.5 \times$ dimension while smaller ranks lead to worse final loss, and we apply them to all multi-head attention layers and feed-forward layers in the models.

- Adam-mini (Zhang et al., 2024b): We use the same hyperparameter as AdamW including $\beta_1 = 0.9$, $\beta_2 = 0.95$, $\epsilon = 10^{-8}$.

### 5.1.1 COMPARISON WITH EXISTING MEMORY-EFFICIENT OPTIMIZERS

As demonstrated in the Fig 3, the loss curves of Adapm closely resemble those of AdamW in both GPT2 series and Llama series, while alternative methods exhibit slower convergence characteristics. We report the memory cost and GPU hours in Tab 2, where the batch sizes per GPU are maximized for each algorithm within GPU memory limits. By implementing a rank reduction to 5% of the original matrix dimensionality, our approach successfully reduces momentum memory consumption to approximately 55% of baseline requirements. Thanks to the memory cut-down, Adapm can support larger batch sizes per GPU. We repeated the experiment five times, confirming that our method possesses stability and reproducibility. Using the parameter settings we provided, the results of our study can be replicated.

### 5.1.2 ABLATION STUDY AND SENSITIVITY ANALYSIS

We conduct experiments on the GPT-2 124M to evaluate the impact of bias correction in Adapm. All configurations share identical hyperparameters, with the sole distinction being the inclusion/exclusion of bias correction in low-rank gradient covariance estimation.

The experimental results in Fig 4(a) demonstrate that omitting bias correction significantly slow down the convergence speed by about 1.96 times even with a rank of $r = 50\%$. The convergence speed decreases definitely compared to bias-corrected method, with training loss plateauing at higher values throughout the optimization trajectory, and low-rank approximations without proper bias correction fail to maintain the original model's convergence properties.

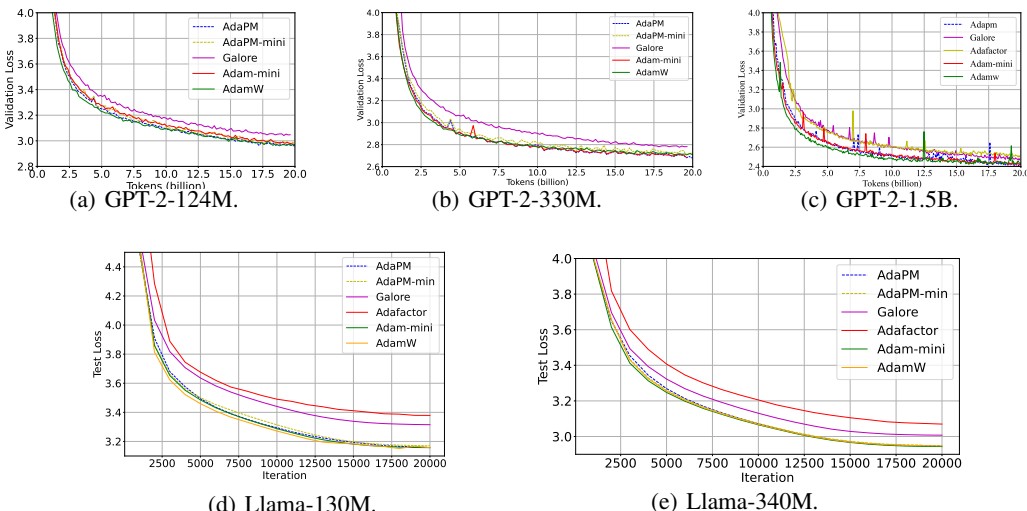

Figure 3: (a)-(c): Loss curves of pre-training GPT-2 series from 124M to 1.5B. (d)(e): Test loss of pre-training Llama-2 series from 130M to 340M. Adapm performs on par or better than AdamW, while other methods perform worse.

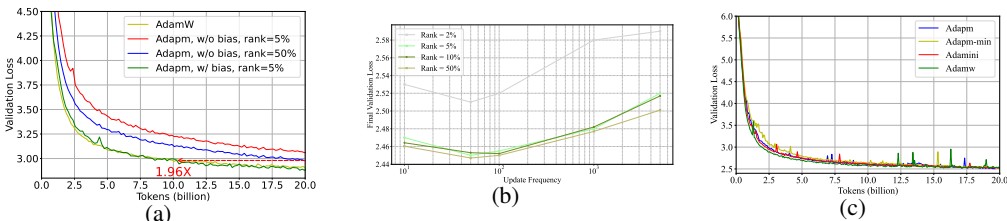

Figure 4: (a) Loss curves of pre-training GPT-2 series with or without bias-correction. (b)Applying Adapm for pretraining GPT-2-1.5B with different rank and update frequency $T$. (c) Applying Adapm to Adam-mini for pretraining GPT-2-1.5B.

We present the final validation loss corresponding to rank ratios of $r = 2\%, 5\%, 10\%, 50\%$, and update frequencies of $T = 10, 50, 100, 1000, 5000$ in Fig. 4(b). The closely overlapping trajectories observed across all tested configurations indicate that our algorithm is robustly insensitive to the choice of rank ratio. Notably, an update frequency of $T = 100$ proves sufficient to maintain competitive convergence speed while minimizing computational overhead. This observed stability suggests that even relatively aggressive low-rank approximations—such as those using only 2% or 5% of the full rank—can effectively preserve the essential optimization dynamics of the model. Moreover, such approximations yield substantial memory savings, highlighting the practical efficiency and scalability of the proposed method in resource-constrained settings.

### 5.1.3 APPLYING ADAPM TO OTHER OPTIMIZER

(maybe show algorithm in appendix) By integrating Adapm with Adam-mini, we develop a memory-efficient optimization approach that simultaneously reduces the memory footprint of both first-order momentum and second-order variance terms in Adam-type optimizers. As shown in Fig 3(c) and Table 2, this combined strategy achieves approximately $95\%$ memory reduction for the optimizer states while maintaining comparable convergence speed to standard Adam. Experimental results in Fig 4(c) demonstrate that this unified approach maintains model performance on pretraining GPT-2-1.5B while dramatically decreasing memory overhead.

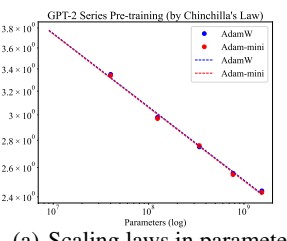
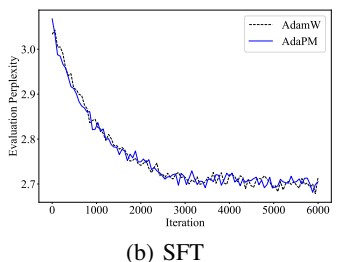
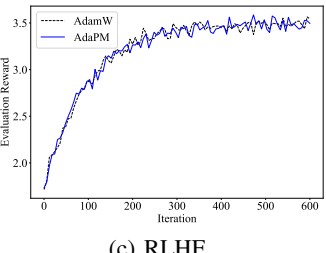

(a) Scaling laws in parameters      (b) SFT      (c) RLHF

Figure 5: Scaling laws in terms of parameters in (a) suggest that Adapm can be scaled up to larger models (if the scaling law holds). (b)(c): SFT,and RLHF when aligning Llama3-8B. AdaPM maintains similar evaluation perplexity and reward to AdamW with 43% less memory.

### 5.1.4 SCALING LAW OF ADAPM

We conduct systematic experiments across the GPT-2 model family (40M, 125M, 350M, 774M, and 1.5B parameters) on OpenWeb-Text dataset to evaluate Adapm's scalability. The results plotted in Figure 5(a) reveals that the log-linear scaling law suggests this optimization approach remains viable for models beyond 1.5B parameters if the scaling law holds, with projected memory savings of 45%. Crucially, the convergence stability shows no degradation with model size, indicating preserved training dynamics.

### 5.2 LLM FINETUNING

We conducted a comprehensive evaluation of AdaPM on both Supervised Fine-Tuning (SFT) and Reinforcement Learning from Human Feedback (RLHF) tasks. Our experiments were based on the Llama3-8B pretrained model. For the SFT phase, we utilized the UltraFeedback dataset (Cui et al., 2023) for training. For the RLHF phase, we implemented the established RLHF pipeline following the methodology described in (Ouyang et al., 2022), with a specific adaptation: we employed ReMax (Li et al., 2023) as our reinforcement learning optimizer. ReMax was chosen as a memory-efficient alternative to the commonly used Proximal Policy Optimization (PPO) algorithm (Schulman et al., 2017), which helps in optimizing the policy towards the preference reward model more efficiently.

The results, as illustrated in Figure 5, demonstrate that AdaPM delivers performance that is comparable to or surpasses that of the AdamW optimizer across both SFT and RLHF benchmarks. This consistent performance highlights AdaPM's effectiveness and robustness in complex training scenarios involving large language models, suggesting its potential as a competitive alternative for modern LLM training pipelines.

## 6 CONCLUSION

We propose AdaPM, an adaptive partial momentum strategy for pretraining and finetuning. We first show that the momentum in transformers optimizer also remain highly redundant, therefore assigning different momentum designs to different blocks. We further improve the partial momentum by a bias-corrected approach using error-feedback technique. AdaPM attains a remarkable 94% momentum memory saving on GPT-2 1.5B without sacrificing convergence. AdaPM can further reduces memory by up to 95% in optimizer states by combining the memory-efficient technique on the second-order statistic.

We note that there remains substantial potential for enhancing the design of AdaPM: explore the partition approach on training of various models such as diffusion models (Ho et al., 2020) and apply the bias-corrected method to the low-rank estimation of activation, which will further reduce the memory cost of training. We leave the development of stronger designs as a future direction.

# 7 ETHICS STATEMENT

Our paper complies with the ICLR Code of Ethics.

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

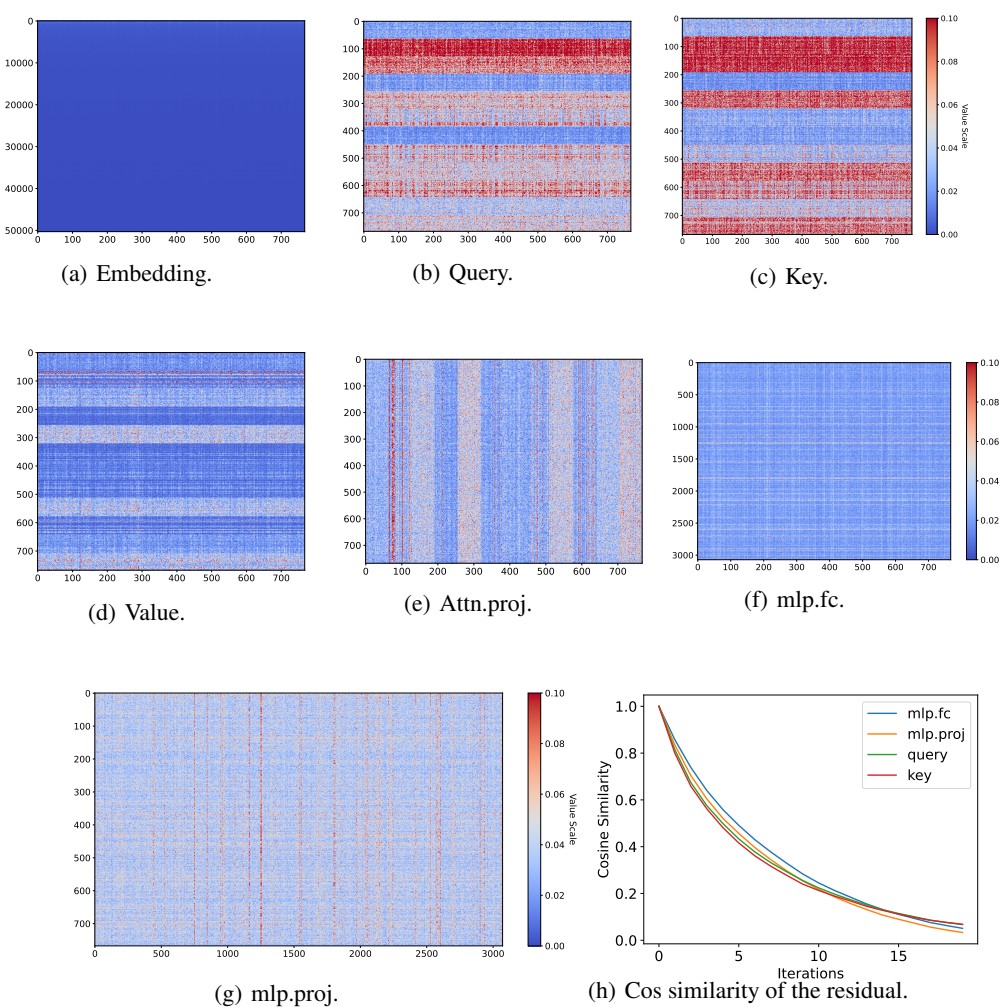

(a) Embedding.  (b) Query.  (c) Key.

(d) Value.  (e) Attn.proj.  (f) mlp.fc.

(g) mlp.proj.  (h) Cos similarity of the residual.

Figure 6: Heatmap of the ratio between gradients and the maximum value in gradient matrices in GPT-2 124M at $10\%$ training step.

## A APPENDIX

### A.1 THE EMPIRICAL INSIGHTS ON SPARSITY OF THE MOMENTUM

