# OpenReview forum: "AdaPM: a Partial Momentum Algorithm for  LLM Training"
_ICLR.cc/2026/Conference — Submitted to ICLR 2026_

### Official Review · Reviewer_c8Nh · 2025-10-31

**Soundness:** 3
**Presentation:** 2
**Contribution:** 2
**Rating:** 2
**Confidence:** 4

**Summary:**

The paper proposes an optimizer named AdaPM that aims at eliminating most of the first order momentum of Adam. It uses a block-aware approach, where momentum is completely removed from the Embedding and Attention Output Projection blocks, maintained for Value blocks and compressed to low-rank for the rest of the blocks. It demonstrates that such an approach could result to comparable performance to Adam at reduced memory.

**Strengths:**

I find the main contribution and novelty of this paper to be the block-aware approach. Table 1. results are interesting, especially the finding that doing low-rank for the Value block has a larger negative impact on the final loss.

**Weaknesses:**

Overall the method is limited in novelty, since low-rank and stateless alternatives to Adam have been already explored extensively in related literature.

Specifically, low-rank Adam that incorporates the error residual has already been proposed and shown to be effective both for first and second order momentum in [1]. I find this very similar to "Approximation residual/Bias correction" step in Algorithm 1, although it is listed as one of the main contributions of this paper.

Also [2] shows results comparable or better than Adam by compressing its both its states up-to rank-1. Near-stateless optimizers have also been proposed as an alternative to Adam and can eliminate most of the first-order momentum for LLM pretraining [3,4]. However the paper only focuses on Adafactor, Galore and Adam-mini as related work, which are older and weaker methods than the state-of-the-art. All these recent related works need to also be considered.

Finally, although the empirical observation about Value blocks is interesting, the paper does not explain why the Value block needs full momentum. Figure 6 also shows that the Value block is sparse too, so this makes the explanation that momentum is redundant due to gradient sparsity not entirely convincing.

Minors:

5.1.3: "(maybe show algorithm in appendix)" is probably forgotten from the draft.

Ref:

[1] Fira: Can We Achieve Full-rank Training of LLMs Under Low-rank Constraint? 2024.

[2] APOLLO: SGD-like Memory, AdamW-level Performance. 2024.

[3] SWAN: SGD with Normalization and Whitening Enables Stateless LLM Training. 2024.

[4] A Minimalist Optimizer Design for LLM Pretraining. 2025.

**Questions:**

1. What numerical precision was used for each experiment? Figure 1 (b) mentions the memory cost in BF16, was solely BF16 used for training?
2. Could the authors share the training code and include all required scripts to test the methods? This would greatly help with reproducibility.

---

### Official Review · Reviewer_NDaA · 2025-11-01

**Soundness:** 3
**Presentation:** 1
**Contribution:** 2
**Rating:** 4
**Confidence:** 4

**Summary:**

This paper proposes an algorithm AdaPM for memory-efficient LLM training. Previous works mainly focus on reducing the memory cost of the second-order momentum of optimizer states, while those works that attempt to reduce the memory cost of the first-order momentum do not avoid the loss of performance. This work first provides a theoretical insight that higher momentum may augment the negative effect of the gradient variance. Then, it shows that the gradient information concentrates on a low-rank structure, motivating the utilization of the low-rank subspace. The above insights lead to the design of AdaPM. It divides the blocks of transformer models into three categories: the first does not adopt momentum, the second adopts the common momentum, and the last updates the parameters with low-rank momentum with correction.

This work conducts the experiments of LLM pretraining and fine-tuning. Benefiting from reducing the memory cost of momentum and the low-rank structure, AdaPM significantly reduces the GPU memory of Adam while maintaining the validation losses. Incorporating AdaPM with Adam-mini could further reduce the memory cost.

The main contribution of this work is that it addresses the dilemma of the high memory cost of the first-order momentum. The experimental results verify this point.

**Strengths:**

This work proposes to divide the blocks in transformer models into three categories and assign different momentum strategies for them. This method represents the originality of this paper. Both theoretical and empirical insights are provided before proposing the algorithm, thus the motivation for designing such an algorithm is clear. The experiments cover several model sizes and both the pretraining and fine-tuning settings. The ablation study and scaling law are also shown in the paper. The experiments are sufficient to reveal the effectiveness.

**Weaknesses:**

This work shows the properties of sparse gradients and low-rank structure in the Section 3.2. Then, it adopts the low-rank update for one category of blocks. However, the above insights and ideas have been observed or proposed in several works, such as GaLore.

The presentation and writing of this work are rather poor. There are many typos and non-standard presentations. For instances, “on the secondorder statistic” in Abstract; “We propose … AdaPM, which reveals that the second-order …” in Introduction, where “second-order” should be “first-order”; “Mermory-efficient” in Introduction; “Adam’s v” and “Adam-mini (Zhang et al., 2024b) partitions the parameters into blocks a proposed principle on Hessian structure” in “Related Work”. There are also some similar problems not listed here. The writing of this paper needs to be checked carefully.

**Questions:**

In lines 179-182, it is stated that “The logic of sparse gradient’s effect on momentum can be summarized as: Low frequency of the gradient signals disrupts gradient accumulation across iterations since most parameter coordinates exhibit gradient signals only rarely, with substantial iteration gaps between occurrences”. Could you please give a more detailed explanation for this point?

In lines 268-269, it is stated that “We apply gradient descent and warm-start from the previous estimate”. How many iterations of gradient descent are implemented in the experiments?

---

### Official Review · Reviewer_92Xz · 2025-11-05

**Soundness:** 2
**Presentation:** 1
**Contribution:** 2
**Rating:** 2
**Confidence:** 3

**Summary:**

This paper introduces AdaPM, a memory-efficient optimizer designed to reduce the high memory cost associated with first-order momentum in optimizers like Adam, particularly for training LLMs. The core idea is that momentum is not equally important across all Transformer components, so AdaPM assigns different strategies: no momentum for blocks with sparse gradients (Embedding, Attention Output), full momentum for blocks that require it (Value), and a bias-corrected low-rank momentum for others (Query, Key, MLP). Experiments claim this approach reduces momentum memory about 50% and, when combined with second-order state reduction (Adam-mini), saves up to 90% of total optimizer state memory. This is achieved while maintaining comparable performance to AdamW on pre-training (GPT-2 up to 1.5B), supervised fine-tuning, and RLHF tasks.

**Strengths:**

1.The paper addresses a very practical and important problem. The high memory cost of the optimizer state is a major bottleneck for training large models. This work focuses on compressing the first-order momentum state, which uses a lot of memory but has not been compressed as much as the second-order state.

2.The key strength is the very strong empirical results. The method achieves large memory savings, reducing momentum memory about 50%. When combined with Adam-mini, the total optimizer state memory is reduced by up to 90%, which gives a big improvement.

3.The authors show strong and comprehensive experiments tasks. They test their method on different model sizes (up to 1.5B) and types (GPT-2, Llama). They also show it works well on different training tasks, including pre-training, SFT, and RLHF.

**Weaknesses:**

Motivation and Justification Part:

1. The theoretical justification in Section 3.1 is bit confusing. It is not clear why Equation (1) is used to analyze validation loss. The paper discusses that momentum may enlarge Term A while improving Term B, but it wasn't fully clear what the net effect on the total validation loss (A+B) is presumed to be. This made the theoretical argument for why momentum should be selectively applied less impactful. Additionally, the analysis in Theorem 1, which seems central to this argument, uses variables 'a' and 'b' that are not defined, making it somehow hard to follow.

2. The claim that sparse gradients are "disrupted" by momentum is not well-justified. The exponential moving average in momentum is designed to let old information decay, so it is unclear why this is a unique problem for sparse gradients. This argument also seems to contradict the use of a second-order momentum in their algorithm, which should also be sparse.

3. The claim of a low-rank gradient structure was supported by an analysis at 10% of the training steps . As gradient properties can evolve, it would be more convincing to see if this low-rank property holds at other key points during training.

Methodology and Algorithm Description:
1. There are a few points in the algorithmic description that could be clarified:
a. There appears to be a discrepancy between the definition of the residual $r_t$ in Algorithm 1 (line 6) and in Equation (3). For example, , Equation (3) uses $L_t R_t$ within the momentum term, while Equation (2) and Algorithm 1 use $L_{t-1}R_{t-1}$.
b. The notation $\tilde{\nabla}f$ is used in several equations but does not seem to be explicitly defined.

2. In Algorithm 1 (line 9), a clip function is used. The purpose of this operation, which is not standard in Adam, was not immediately clear from the text.

Finally, the experimental evaluation is missing important and recent baselines. The paper does not compare against other relevant memory-efficient optimizers like SWAN[1] or SCALE[2], which makes the comparison incomplete.

[1] Chao Ma, Wenbo Gong, Meyer Scetbon and Edward Meeds. SWAN: SGD with Normalization and Whitening Enables Stateless LLM Training. arXiv preprint arXiv:2412.13148

[2] Athanasios Glentis, Jiaxiang Li, Andi Han and Mingyi Hong. A Minimalist Optimizer Design for LLM Pretraining. arXiv preprint arXiv:2506.16659

**Questions:**

1. Theoretical Motivation: Could the authors elaborate on the net effect of momentum on the total loss (Term A + Term B) in Section 3.1? Also, could you please define the 'a' and 'b' variables used in Theorem 1?

2. In Section 3.2, can you please explain why "Low frequency of the gradient signals" is a problem for momentum? The momentum's exponential moving average is already designed to handle decaying history. Also, if this is a problem, why is the second-order momentum (which should also be sparse) still effective?

3. Could the authors please explain the purpose of the clip function in Algorithm 1 (line 9)? What value is it clipping to, and what is its impact on the optimization?

5. Can you comment on how AdaPM would compare against more recent memory-saving optimizers, such as SWAN[1] and SCALE[2]?

[1] Chao Ma, Wenbo Gong, Meyer Scetbon and Edward Meeds. SWAN: SGD with Normalization and Whitening Enables Stateless LLM Training. arXiv preprint arXiv:2412.13148

[2] Athanasios Glentis, Jiaxiang Li, Andi Han and Mingyi Hong. A Minimalist Optimizer Design for LLM Pretraining. arXiv preprint arXiv:2506.16659

---

### Meta-Review · Area_Chair_PmTv · 2025-12-28

**Summary:**

This submission proposes AdaPM, a memory-efficient optimizer that reduces the first-order momentum memory cost of Adam by exploiting block-wise heterogeneity in Transformer architectures. The method assigns different momentum strategies to different blocks, removing momentum for some, retaining full momentum for others, and applying bias-corrected low-rank momentum elsewhere. Across reviewers, there is strong agreement on the practical importance of the problem and the strength and breadth of empirical results. However, the paper is assessed as below the acceptance threshold due to insufficient justification of key claims, incomplete positioning against recent optimizers, and weak presentation quality. In particular, reviewers found that the theoretical arguments motivating selective momentum usage are confusing, and that the novelty relative to recent low-memory or stateless optimizers is not convincingly established.

**Reviewer Concerns:**

All concerns are not addressed.

**Reviewer Scores:**

Unchanged.

---

### Decision · Program_Chairs · 2026-01-26

Reject